# Are There Any Cognitive and Behavioral Changes Potentially Related to Quarantine Due to the COVID-19 Pandemic in People with Mild Cognitive Impairment and AD Dementia? A Longitudinal Study

**DOI:** 10.3390/brainsci11091165

**Published:** 2021-08-31

**Authors:** Marianna Tsatali, Despina Moraitou, Eleni Poptsi, Eleni Sia, Christina Agogiatou, Moses Gialaouzidis, Irene-Maria Tabakis, Konstantina Avdikou, Evaggelia Bakoglidou, Georgia Batsila, Dimitrios Bekiaridis-Moschou, Ourania Chatziroumpi, Alexandra Diamantidou, Andromachi Gavra, Eleni Kouroundi, Despina Liapi, Nefeli Markou, Fani Ouzouni, Chrysa Papasozomenou, Aikaterini Soumpourou, Magdalini Tsolaki

**Affiliations:** 1Greek Association of Alzheimer’s Disease and Related Disorders (GAADRD), 54643 Thessaloniki, Greece; despinamorait@gmail.com (D.M.); poptsielena@gmail.com (E.P.); eleana.sia@yahoo.com (E.S.); ag.christina@gmail.com (C.A.); moses_gf@hotmail.com (M.G.); irma.tabakis@gmail.com (I.-M.T.); avdikoukonstadina@yahoo.gr (K.A.); litsa.bak@gmail.com (E.B.); gewr_gia@yahoo.gr (G.B.); d_bekiaridis@hotmail.com (D.B.-M.); rchatziroubi@gmail.com (O.C.); a.diamantidou@alzheimer-hellas.gr (A.D.); antagavra@gmail.com (A.G.); kouroundi@alzheimer-hellas.gr (E.K.); liapides84@gmail.com (D.L.); markou.nef1@gmail.com (N.M.); fanie_ouz@yahoo.com (F.O.); chrysa.kav@gmail.com (C.P.); katerinasoum@gmail.com (A.S.); tsolakim1@gmail.com (M.T.); 2Laboratory of Neurodegenerative Diseases, Center for Interdisciplinary Research and Innovation (CIRI-AUTh) Balkan Center, Buildings A & B, Aristotle University of Thessaloniki, 57001 Thessaloni, Greece; 3Lab of Psychology, Section of Experimental & Cognitive Psychology, School of Psychology, Aristotle University of Thessaloniki, 54124 Thessaloniki, Greece; 41st Department of Neurology, School of Medicine, Faculty of Health Sciences, Aristotle University of Thessaloniki, 54124 Thessaloniki, Greece

**Keywords:** mood status, coronavirus-19 pandemic, cognitive status, effect of quarantine in older adults, everyday functioning

## Abstract

The aim of the study was to examine potential cognitive, mood (depression and anxiety) and behavioral changes that may be related to the quarantine and the lockdown applied during the COVID-19 pandemic in Greek older adults with mild cognitive impairment (MCI), and AD dementia in mild and moderate stages. Method: 407 older adults, diagnosed either with MCI or AD dementia (ADD), were recruited from the Day Centers of the Greek Association of Alzheimer Disease and Related Disorders (GAADRD). Neuropsychological assessment was performed at baseline (at the time of diagnosis) between May and July of 2018, as well as for two consecutive follow-up assessments, identical in period, in 2019 and 2020. The majority of participants had participated in non-pharmacological interventions during 2018 as well as 2019, whereas all of them continued their participation online in 2020. Results: Mixed measures analysis of variance showed that participants’ ‘deterioration difference—D’ by means of their performance difference in neuropsychological assessments between 2018–2019 (D1) and 2019–2020 (D2) did not change, except for the FUCAS, RAVLT, and phonemic fluency tests, since both groups resulted in a larger deterioration difference (D2) in these tests. Additionally, three path models examining the direct relationships between performance in tests measuring mood, as well as everyday functioning and cognitive measures, showed that participants’ worsened performance in the 2019 and 2020 assessments was strongly affected by NPI performance, in sharp contrast to the 2018 assessment. Discussion: During the lockdown period, MCI and ADD patients’ neuropsychological performance did not change, except from the tests measuring verbal memory, learning, and phonemic fluency, as well as everyday functioning. However, the natural progression of the MCI as well as ADD condition is the main reason for participants’ deterioration. Mood performance became increasingly closely related to cognition and everyday functioning. Hence, the role of quarantine and AD progression are discussed as potential factors associated with impairments.

## 1. Introduction

According to the declaration of the World Health Organization in March 2020, severe acute respiratory syndrome coronavirus 2 (SARS-CoV-2), the main cause of the subsequently named coronavirus disease 2019 (COVID-19), constitutes a viral infectious disease initially emerging in Wuhan, China, in late 2019 [1]. SARS-CoV-2 was spread worldwide and, therefore, many countries have posed national quarantine. In more detail, COVID-19 constitutes a respiratory illness with high heterogeneity in symptoms’ prevalence [2]. As expected, mortality rates are higher in the older adult population and those with comorbid health problems, including people living with dementia (PwD) [3].

Due to the existence of the pandemic and subsequent lockdown measures, which is a term conventionally used to describe the mass quarantine imposed, various safety strategies have been implemented by almost the majority of government authorities, resulting in both positive and negative consequences. Although social isolation and restriction were associated with reduced COVID-19 transmission, the scientific community as well as previous research [4] highlight the severe negative effects of quarantine on mental health among young and older adult populations. More specifically, recent data [5] showed that prolonged homestay of older adults may be accompanied by deleterious side effects, which can impact physical as well as mental health, resulting in deteriorating quality of life indices. Due to the widespread quarantine measures applied, post-traumatic stress disorder as well as depressive symptomatology was observed in a large percentage of a research cohort [4]. Current observations support that the COVID-19 pandemic can lead to a possible outbreak of psychiatric disorders [6,7], due to various factors such as fear and agony, sense of loneliness due to social distancing, uncertainty about the future, confusion, anger about the pandemic’s duration, as well as a feeling of being exposed and stigma in the case of being infected by the virus.

People with dementia (PwD), considered as vulnerable members of society [8], might be severely affected by the general home confinement ‘lockdown’. The main reason is attributed to the fact that those patients are strongly dependent on various health services and social support; the majority of them visit local dementia day care centers in order to participate in various activities. Thus, given the condition in which these patients were isolated due to the pandemic, it can be assumed that they were somewhat affected in the aforementioned domains, without always being able to express their feelings, anxieties, worries, and ask for support. 

Another significant parameter in PwD is coexisting neuropsychiatric disorders, which, combined with the psychological consequences due to the general lockdown [9], as well as social isolation and deprivation [8], create a vicious cycle, leading to a growing increase in psychiatric disorders in patients with dementia, which also affect their caregivers. According to recent research data [10,11,12], isolation and social distancing are closely associated with poor mental health, e.g., depressive symptomatology, cognitive deterioration, as well as physical decline and finally, high mortality. Furthermore, people with milder forms of dementia and MCI [8] may also have difficulties in complying with the security measures due to mood disorders such as depression and apathy. 

The aforementioned situation is further aggravated by the fact that PwD did not have access to physical exercise, physiotherapy services and the chance of frequent outings, resulting in a plethora of negative consequences both on their physical as well as mental health. Moreover, limited cognitive stimuli could definitely lead to cognitive deterioration in these people who have already existing cognitive decline. To sum up, lack of physical support combined with reduced social support and lower levels of cognitive stimulation can result in increasing possibility of behavioral disorders, such as delirium and irritability [9], as well as cognitive decline [8,13]. Additionally, according to recent data [14], the duration of the confinement measures was positively correlated with neuropsychiatric symptoms’ severity and subsequently, this increase was also related to reduced cognitive function in Alzheimer’s Disease Dementia (ADD) patients. Despite the fact that one can hypothesize that the general lockdown had significant repercussions in older adults and those with minor and major neurocognitive disorders, until now, the extent of this phenomenon is unclear. Additionally, to our knowledge, there are no previous longitudinal studies using neuropsychological evaluation to measure the extent to which those with MCI as well as ADD were affected by means of their neuropsychological performance due to the pandemic. Therefore, this study aims to shed light on whether the COVID 19 pandemic affected neuropsychological performance in the aforementioned populations. This is quite important because it is crucial to identify whether these groups are at greater risk of cognitive deterioration during the pandemic and find out possible protective factors. 

### Aim of the Study

The researchers aimed at identifying whether people with MCI as well as those with ADD (mild and moderate stages) experienced significant changes at a cognitive, emotional and behavioral level, due to COVID-19-related lockdowns. 

Specifically, there was the hypothesis (1) that MCI and ADD patients in quarantine (1a) could experience cognitive and functional decline, (1b) as a result of emotional and behavioral changes during the lockdowns, in comparison with their status as assessed one and two years before.

## 2. Methods

### 2.1. Participants

In the current study, we recruited participants from the Greek Association of Alzheimer’s Disease and Related Disorders (GAADRD) and, specifically, the Day Care Centers of “Saint John” (DCCSJ) and ‘’Saint Helen’’ (DCCSH) in Thessaloniki. GAADRD provides diagnostic services, pharmacological and non-pharmacological interventions for older adults with mild to severe cognitive deficits. The sample consisted of two different groups: (a) people with MCI (amnestic and non-amnestic); (b) people with the diagnosis of ADD according to the criteria of the Diagnostic and Statistical Manual of Mental Disorders, fifth edition (DSM-5) [15]. They had initially come to the two Day Care Centers from May to July in 2018, in order to undergo a neuropsychological and neurological evaluation due to their subjective and objective memory complaints. After receiving the diagnosis in 2018, participants from both groups came to the Day Care Centers in 2019 and 2020, the same three-month period (May–July), similarly with their initial testing, to do their follow-up evaluations. 

In more detail, participants followed the diagnostic protocol of the GAADRD, which involves neurological, neuropsychiatric and neuropsychological assessment, neuroimaging (CT scan or magnetic resonance imaging), as well as blood tests to exclude other types of reversible cognitive impairment or dementia. Along with the neuropsychological tests, psychometric tools that measure affect and behavioral problems were also administered. The official neuropsychological assessment was administered by the neuropsychologists who constitute the expert group of the GAADRD. 

During 2018 and 2019, as well as the first two months of 2020, all of the study’s participants had medical support from expert neurologists, when needed, and also participated in the non-pharmacological interventions provided by the Day Care Centers. To become more descriptive, different programs of cognitive training, involving a broad area of cognitive functions such as attention (visual–auditory, sustained–divided), language (naming and fluency) and aspects of executive function (working memory, inhibition, shifting, cognitive flexibility), were implemented during this time period. The concept is that different stimuli and tasks, as well as new experiences, are offered to people with MCI and dementia, in order to boost their brain function.

During the first lockdown the participants continued receiving psychosocial and medical support via Skype, Zoom and Viber platforms, according to their needs. More specifically, they were encouraged to contact their doctor in order to help them with prescriptions or any other medical issue. Additionally, through the digital platforms as well as the telephone online service, the beneficiaries were able to receive medical assistance, psychosocial support or any other support in any urgent issue that could have arisen during the pandemic. This practice is in accordance with other medical centers across America and Europe, which also had adapted teleconsultation services [16,17].

From the initial population of people who underwent the official diagnostic protocol in 2018 (*n* = 3.396), 470 had two follow-up assessments in two consecutive years, that is, in 2018 and 2019. Just after the restrictive measures (2020), these participants were asked to undergo the neuropsychological evaluation again, and finally, 407 of them responded to that request and were included in the study, having 3 consecutive assessments. 

Participants with MCI were diagnosed by the DSM-5 criteria for Mild Neurocognitive Disorders, and those with ADD in mild and moderate stages followed the DSM-5 criteria for Major Neurocognitive Disorders. Specifically, the inclusion criteria for MCI were: (a) diagnosis of MCI according to the DSM-5 [15], (b) Mini Mental State Examination (MMSE) total score ≥ 26, (c) stage 3 of the disease according to the Global Deterioration Scale [18], (d) 1.5 standard deviation (SD) below the normal mean according to age and education, in at least one cognitive domain according to the utilized neuropsychological tests. The inclusion criteria for ADD were: (a) diagnosis of Alzheimer’s disease according to the DSM-5, (b) MMSE total score ≤ 23 and ≥ 12, (c) stage 4 and 5 of the disease according to the Global Deterioration Scale [18], (d) 2 standard deviations (SD) below the normal mean according to age and education, in at least one cognitive domain according to the utilized neuropsychological tests as well as subsequent deterioration in independent living, (e) those who had ADD in severe stage were excluded, based on Criterion (a).

The exclusion criteria in both groups are the following: (a) history of psychiatric illness such as Schizophrenia or mood disorder (Major Depression–General Anxiety Disorder), (b) substance abuse or alcoholism, (c) history of traumatic brain injury, (d) neurological disorders such as hydrocephalus, Parkinson’s disease, encephalitis, brain tumor, epilepsy and stroke history, (e) thyroid, (f) diabetes, (g) drug treatment with opioids, B12, folate, thyroid, and (h) severe sensory deficits. It is worth noting that older adults with mild cardiovascular problems such as mild hypertension were not excluded in the two groups of our sample.

### 2.2. Procedure

Every neuropsychological evaluation lasted for two hours, divided into two different face to face sessions, in order to obtain the participants’ best performance and reducing the possibility of becoming tired. Initially, the following screening tests, adapted in the Greek population, were administered: Mini Mental State Examination (MMSE; Greek cut-off scores from Fountoulakis et al. [19]), and Montreal Cognitive Assessment (MoCA; Greek cut-off scores from Poptsi et al. [20]). Short-term, long-term verbal memory and verbal learning ability were measured by Rey’s Verbal Learning Test (RAVLT, three variables; RAVLT first trial measures short-term memory, RAVLT fifth trial measures verbal learning, and RAVLT delayed recall measures long-term memory; Greek cut-off scores from Messinis et al. [21]). Verbal fluency was measured by the phonemic verbal fluency test adapted in Greek by Kosmides et al. [22]. Visuospatial abilities were measured by the RCFT, specifically Copy as well as Delayed Recall trials, which was adapted by Tsatali et al. [23]. Finally, the Wechsler Adult Intelligence Scale (WAIS) Digit symbol (adapted in Greece by Tsatali et al. [24]) subtest was administered to measure processing speed. Additionally, independent living was measured by the Functional Cognitive Assessment Scale (FUCAS; Greek cut-off scores from Kounti et al. [25]). The Short Anxiety Screening Test (SAST) [26,27] was used in order to measure mood deficits. Finally, the Greek Neuropsychiatric Inventory (NPI) was also conducted in one family member of participants with Alzheimer’s disease dementia, so as to measure the level of neuropsychiatric symptoms [28]. In more detail, NPI measures behavioral and mood disturbances and specifically, hallucinations/delusions, irritability/aggression, depressive and anxiety symptoms, inappropriate behaviors, and psychomotor delay or agitation.

### 2.3. Ethics

All study participants read the information sheet and signed the informed consent during the initial clinical visit, stating that the research group of the GAADRD have the permission to use their demographic data, which would be anonymized, such as gender, age, and education, as well as their performance in the neuropsychological tests, for research purposes. For PwD in mild and moderate stages, a legal representative read the consent and signed the relevant document.

The study was approved by the Scientific and Ethics Committee of the GAADRD (Scientific Committee Approved Meeting Number: 60/14-10-2020), which follows the new General Data Protection Regulation (EU) 2016/679 of the European Parliament and of the Council of 27 April 2016 on the protection of natural persons with regard to the processing of personal data and on the free movement of such data, as well as the principles outlined in the Helsinki Declaration.

### 2.4. Data Analysis

The statistical analysis was initially performed by SPSS software version 25 (IBM; SPSS Statistics for Windows, Version 25.0. Armonk, NY: IBM Corp 25.0). In order to examine whether there were any significant differences between the groups and ‘deterioration differences’ related to times of assessment, mixed measures ANOVA (2 (group: people with MCI, people with ADD) × 2 (deterioration difference—D: D1 = deterioration difference between 2018 and 2019 assessments; D2 = ‘deterioration difference’ between 2019 and 2020 assessments)) was used. Regarding the variable of ‘deterioration difference’, the need to insert this variable was based on the rationale that there were many measures and a large amount of information, and our aim was to focus on some of them that clearly and strongly deteriorate during the period of interest. D1 ‘deterioration difference’ was calculated by subtracting participants’ performance in 2019 from their performance in 2018, whereas the D2 was created by subtracting participant’s performance in 2020 from their performance in the 2019 neuropsychological assessment. Hence, the higher the D indices, the higher the deterioration of performance as regards most tests. However, in the NPI and FUCAS, a higher score describes worsening performance and, therefore, the lower the D indices, the higher the deterioration was. 

In regard to neuropsychological tests administered only in the MCI group in the 2018, 2019 and 2020 assessments, repeated measures ANOVA was applied to the respective data, using the deterioration difference as a “factor” (D: D1 = deterioration difference between 2018 and 2019 assessments; D2 = ‘deterioration difference’ between 2019 and 2020 assessments). 

Moreover, because of the expanded neuropsychological assessment, the Bonferroni correction was utilized, setting the level of significance at α = 0.004 (0.05/number of tests-variables). 

In order to examine the direction of relationships between mood variables and cognitive and everyday functioning variables in the three neuropsychological assessments, path analysis was conducted in EQS (version 6.1) statistical software [29]. A maximum likelihood estimation procedure was performed. Regarding the confirmation of a path model, a non-significant level of Goodness of Fit index χ^2^, that is *p* > 0.05, is indicative of a good fit of the model to the data. In addition, when the value of Root Mean Square Error of Approximation (RMSEA) is <0.05, it is also an indication of the good fit of the model to the data. RMSEA values ranging from 0.06 to 0.08 indicate a reasonable and, therefore, acceptable approximation error. Comparative Fit Index (CFI) examines whether the data fit a hypothesized path model compared to the basic model. Values greater than 0.90 indicate adequate fit of the model to the data, whereas values close to 1.00 indicate a good fit [30]. Moreover, to improve model fit, we examined the modification indices, namely the Wald and the Lagrange tests, which represent frequently used statistics to identify focal areas of a misfit in a path analysis solution [30].

## 3. Results

### 3.1. Descriptive Statistics

In detail, (a) 296 older adults with MCI (99 men and 197 women, age range: 50 to 92 years, M = 71.19, SD = 7.73, education range: 2 to 20 years, M = 12.38, SD = 3.85) and (b) 111 older adults with ADD in mild and moderate stages (47 men and 64 women, age range: 51 to 90 years, M = 77.32, SD = 3.85, education range: 2 to 18 years, M = 9.28, SD = 4.89) took part in the current study. It must be mentioned that none of the participants had received a laboratory-confirmed diagnosis of COVID 19, after being asked by the doctor of the GAADRD.

No significant differences were found between the two groups by means of gender (χ(1) = 2.913, *p* = 0.088). However, there were statistically significant differences between the two groups in demographic characteristics such as age (F(1, 372) = 46.57, *p* < 0.001), and education (in years) (F(1, 372) = 41.41, *p* < 0.001). Therefore, older adults with MCI were more educated compared to ADD who were older and less educated. 

The study’s sample characteristics are presented in Table 1. 

The mean and standard deviation of all tests for all three assessments are given in Table 2, together with their significant differences.

### 3.2. The Effects of Diagnostic Group and ‘Deterioration Difference’ on Cognition, Everyday Functioning, Affect and Behavior

#### 3.2.1. General Cognitive Status (MMSE, MoCA)

Regarding the MMSE, a higher score value is related to a higher level of general cognitive status. Hence, in regard to the deterioration difference, the higher the value of the mean score, the lower the level of cognitive status. The analysis showed that there was a main effect of diagnostic group, that is, MCI participants had a lower deterioration rate in MMSE scores (M = 0.013, SΕ = 0.093) than the ADD group (M = 1.624, SΕ. = 0.152) as expected. 

Repeated measures analysis of variance (ANOVA) showed that the MoCA screening test score did not differ across the two deterioration difference variables in the MCI group. Data from the ADD group were not extracted because MoCA was not administered in this population.

#### 3.2.2. Activities of Daily Living—Everyday Functioning (FUCAS)

Concerning the FUCAS test, the higher the mean value, the lower the level of everyday functioning. A main effect of deterioration difference was found, with FUCAS D1 (M = −1.32, SE = 0.250) denoting a higher functioning level than FUCAS D2 (M = −2.71, SE = 0.280). There was also a significant main effect of group—that is, MCI participants had a lower deterioration difference (M = 0.009, SE = 0.210) than the ADD group (M = −4.04, SE = 0.33) as again expected. The interaction between group and condition was also significant. The subsequent Scheffe post hoc test showed that the D2 value was higher in the ADD group as compared to MCI participants (Ι—J = 4.055, *p* < 0.001). 

#### 3.2.3. Mood and Behavioral Tests (SAST and NPI)

Concerning the anxiety symptoms measured by SAST, no significant interaction or main effects were found. As regards the NPI test, no significant effect of deterioration difference and diagnostic group was found. The interaction effect was also non-significant.

#### 3.2.4. Tests Measuring Short-Term Memory and Visual Perception (RAVLT, 1st Trial; RCFT, Copy Trial)

By means of measuring short-term memory with RAVLT, and specifically, the immediate recall (first trial), there was a main effect of deterioration difference. The RAVLT trial 1 score of the D1 (M = −0.13, SE = 0.13) was lower than that of the D2 (M = 0.93, SE = 0.14). However, no main effect of group as well as group by condition interaction was found.

Neither a significant interaction effect for the RCFT copy trial nor a significant main effect of deterioration difference was found. The only exception is that MCI participants had a lower deterioration difference (M= −0.924, SE = 0.211) than the ADD group (M = 0.19, SE = 0.34) as again expected.

#### 3.2.5. Test Measuring Verbal Learning (RAVLT, 5th Trial)

Regarding verbal learning ability (RAVLT 5th trial), there was a significant main effect of deterioration difference. Verbal learning ability mean deterioration difference in D2 (M = 1.09, SE = 0.14) was higher than this of the D1 (M = −0.01, SE = 0.14). Nevertheless, no main effect of group as well as group by condition interaction was found.

#### 3.2.6. Tests Measuring Visual and Verbal Long-Term Memory (RAVLT, Delayed Recall; RCFT, Delayed Recall)

By means of the delayed recall condition of the RAVLT, there was a main effect of deterioration difference. RAVLT mean deterioration difference in D2 (M = 1.53, SE= 0.18) was higher than this of the D1 (M = 0.03, SE = 0.14). A significant group by deterioration difference interaction was also found. The subsequent Scheffe post hoc test showed that mean deterioration difference in D2 in the MCI group was higher than that of the ADD group (I—J = 0.344, *p* = 0.002). No significant main effect of group was found.

In regard to the RCFT delayed recall, no significant interaction effect as well as main effect of deterioration difference was found. The only exception was the main effect of diagnostic group, according to which MCI participants had a lower RCFT delayed recall mean deterioration difference (M = −1.11, SE = 0.17) than the ADD group (M = 0.43, SE = 0.27), as expected.

#### 3.2.7. Tests Measuring Verbal Fluency and Executive Function

According to the results for the phonemic fluency condition, there was a main effect of deterioration difference, with D2 (M = 0.43, SE = 0.14) being higher than D1 (M = −0.26, SE = 0.14), which means that the deterioration difference worsened. There was also a significant main effect of diagnostic group—that is, MCI participants had a lower mean deterioration difference (M = −0.20, SE = 0.08) than the ADD group (M = 0.36, SE = 0.13), as expected. 

Repeated measures analysis of variance between D1 (M = −1.23, SE = 0.40) and D2 (M = 1.74, SE = 0.39) deterioration difference in the MCI group by means of their performance in DSST showed that the D2 was significantly increased compared to the D1. 

Table 3 shows the indices for main effects and interaction effects (if any) for each test. 

### 3.3. The Directed Relationships between Diagnostic Group, Mood, Behavioral, Cognitive and Everyday Functioning Performance in 2020 (during the Lockdowns), 2019 and 2018 Neuropsychological Assessments

In the next step, in order to find the potential direction of the relationships among diagnosis (MCI, ADD), mood and behavioral symptomatology, and cognitive abilities and everyday functioning (hypothesis 1b), path analyses were performed separately for the 2018, 2019 and 2020 neuropsychological assessments, including only the cognitive variables and the functioning variables that were found to decline significantly in terms of deterioration difference. Therefore, we entered the variables of diagnostic group, of NPI and SAST, and those five variables for which a significant deterioration difference was found for both diagnostic groups in the path analysis for the three consecutive years’ assessments. 

The path model, which was finally confirmed, χ^2^(4, 354) = 12.39, *p* = 0.014, CFI = 0.99, SRMR = 0.02, RMSEA = 07 (90%CI: 0.03–0.12), showed that diagnostic group predicted all other variables, as expected, with MCI patients performing better than the ADD group. However, the most important finding in this model (see Figure 1) was that NPI performance in the 2020 assessment predicted everyday functioning (FUCAS) and all cognitive performances in the expected direction (see Figure 1). The SAST predicted phonemic fluency performance as well. Moreover, the finding that diagnostic group predicted NPI and SAST performance, and in this way, perhaps indirectly predicted everyday functioning and cognitive performances, is indicative that the ADD group is probably more affected by the lockdowns at the mood–behavioral level, than MCI patients, and via this pathway, is more affected at the cognitive and everyday functioning level too (see Figure 1). Thus, this model seems to confirm hypothesis 1b. 

Nevertheless, in order to double check if hypothesis 1b was confirmed, we ran the same path analysis, using the respective variables for 2019 and 2018 assessments separately (see Figure 2 and Figure 3). The path model, which was finally confirmed for 2019, χ^2^(6, 374) = 19.91, *p* = 0.002, CFI = 0.99, SRMR = 0.02, RMSEA = 0.08 (90%CI: 0.04–0.12), was almost similar to that of 2020. The model again showed that diagnostic group predicted all other variables, as expected, with MCI patients performing better than the ADD group. NPI performance in the 2019 assessment predicted everyday functioning (FUCAS) and all cognitive performances in the expected direction (see Figure 2). Only the SAST did not significantly predict phonemic fluency performance in this case. Moreover, the finding that diagnostic group predicted NPI performance, and in this way, perhaps indirectly predicted everyday functioning and cognitive performances, is indicative that ADD patients were probably more affected by the progression of their disease at the mood–behavioral level than MCI patients, and via this pathway, became more affected at the cognitive and everyday functioning level too (see Figure 2). Thus, although similar, in sharp contrast to the path model confirmed for 2020, this model seems to disconfirm hypothesis 1b, since it shows that the “connection” between mood–behavioral variables and cognitive and everyday functioning variables is probably due to the progression of Alzheimer’s disease. 

In the next step, we ran the same path analysis, using the respective variables for the 2018 assessment (see Figure 3). Interestingly, in this case, the path model, which was finally confirmed, χ^2^(10, 309) = 27.80, *p* = 0.001, CFI = 0.98, SRMR = 0.04, RMSEA = 07 (90%CI: 0.04–0.11), showed that as in the aforementioned models, diagnostic group predicted all other variables, as expected, with MCI patients performing better than the ADD group. However, there was no relationship between NPI with everyday functioning (FUCAS) and cognitive performances (see Figure 3). Only the SAST predicted phonemic fluency performance. Thus, in the 2018 assessment, mood and behavioral status was almost not related at all to everyday functioning and cognition. The indirect prediction of diagnosis, via affect and behavior, consequently disappeared. Such a model for the 2018 assessment can be explained by the fact that all patients were measured at an earlier stage of their disease, compared to the next two assessments. It seems that at this stage, mood–behavioral symptoms are not yet related and interact with cognitive decrements and everyday functioning decline.

## 4. Discussion

In the current study, we focused on studying the potential consequences of the lockdown due to the COVID-19 pandemic on the cognitive, functional as well as emotional and behavioral levels of older adults suffering from mild and moderate neurocognitive diseases who had received telehealth support during the pandemic. A possible severe issue in these populations is that the extended lockdowns, applied as a protecting strategy, have a detrimental effect on cognition [31]; hence, older adults with MCI and ADD confined due to the quarantine should be assumed as top priority by means of research as well as enablement of clinical support access. 

Contrary to our initial hypothesis, MCI and ADD patients without a COVID-19 diagnosis did not experience significant mood and behavioral changes during the lockdowns, since the deterioration of the respective scores between 2019 and 2020 did not significantly differ from the deterioration between 2018 and 2019. In line with this, both groups did not manifest increased levels of cognitive or functional decline. Additionally, according to the results, the deterioration difference of participants’ performance in specific tests, namely, the RAVLT test (subtests which measure short-term memory, verbal learning and long-term memory), the phonemic verbal fluency test (subtest which measures phonemic verbal fluency) and also the FUCAS test (which measures executive function in daily life activities) was higher between the 2019 and 2020 assessments, as compared to the same difference between the 2018 and 2019 assessments, mainly in people with ADD. However, the path analyses showed that the deterioration rate in the 2020 assessment can only be attributed to participants’ condition, similarly to those from the two previous years. Consequently, it can be assumed that the natural progression of MCI as well as ADD is the main reason for participants’ deterioration in the aforementioned tests. Finally, it is worth mentioning that the results of the current study cannot be generalized in a cognitively intact older adult population. 

Van Maurik et al. [32] found that older adults with Subjective Cognitive Decline, (SCD), MCI as well as ADD have increased self-reported worries concerning a possible cognitive decline due to the COVID-19 pandemic, which is also in line with recent data from Barguilla et al. [33] that people with MCI and ADD manifested cognitive decline according to their caregivers’ self-reports. Nevertheless, in the aforementioned studies, they did not administer neuropsychological tests to examine whether these worries are associated with clinically proven decline. Finally, a Greek study by Tsapanou et al. [34] found that those with MCI and dementia were affected due to the quarantine in Spring 2020 in various domains such as mood and communication; they also exhibited lower levels of compliance with respect to hygiene safety measures against COVID-19. However, the aforementioned researchers used either self-reported questionnaires or non-validated scales, and therefore, no objective tests of cognition were administered in order to reach solid conclusions. To summarize, to our knowledge, there are no previous data regarding a clinically meaningful effect of quarantine on people with mild and major neurocognitive disorders, as measured by neuropsychological tests used in clinical practice. 

Contrary to our results, previous studies have shown that mood and behavioral disturbances were increased after the quarantine period in Spring 2020. Specifically, a Spanish study that measured the impact of confinement strategies on neuropsychiatric symptoms’ development as well as the quality-of-life indices of 20 PwD and 20 people with MCI [35] showed that according to pre and post quarantine BPSD (Behavioral and Psychological Symptoms of Dementia) assessment, apathy, agitation and aberrant motor behavioral symptoms increased. Additionally, recent data [36] suggest that after one month of lockdown, people with dementia and milder neurocognitive disorders had manifested behavioral disturbances, mainly agitation/aggression, depression and apathy. Nevertheless, in the aforementioned studies, neither of the two research groups conducted a longitudinal study, nor did they use clinically validated tools for the assessment of BPSD symptoms.

The three configural path models, and mainly, the qualitative differences between them, do not seem to confirm our hypothesis (1b), at least in terms of diagnosis and mood–behavioral disturbances’ relationships with aspects of cognition and everyday functioning that worsen between consecutive assessments before and during the lockdown. Hence, one can assume that the cognitive and functional decline, which was found in our study, follow the worsening trajectory seen in MCI as well as ADD, mainly in moderate stages. More specifically, with regard to the relationships between mood, behavioral and cognitive qualities, we had initially hypothesized that any cognitive and functional decline during the lockdowns could be attributed to mood and behavioral disturbances, as a result of the lockdown conditions. However, as Figure 1 and Figure 2 show, we have indeed found that everyday functioning, learning, verbal long-term memory, short-term memory and phonemic fluency—which deteriorated faster between the 2019 and 2020 assessments, compared to those between 2018 and 2019—were predicted by the NPI test both in 2019 and 2020. Therefore, NPI was found to predict deterioration irrespective of the quarantine being imposed. A second interesting finding was that diagnosis (MCI, ADD), besides the direct effects on the variables in the model, had additional indirect effects on cognition and functioning via mood and behavioral measures, as ADD patients experienced a larger decline compared to the MCI group. Running the same path model for the 2018 assessment, no such relationships concerning the predictive ability of NPI and the potential indirect effects of diagnosis via NPI were found. Concerning the SAST test, there were no obvious findings about whether it would possibly predict verbal fluency performance, due to the quarantine, since it was found to significantly predict verbal fluency index in both years, except from 2019. 

From another point of view, verbal abilities as measured by the RAVLT and phonemic fluency test could decline due to negative affect and feeling of loneliness caused by the lockdown condition, which subsequently could lead to decreasing interest and availability to communicate or vice versa; the increased interest to communicate due to the lockdown measures could lead to exacerbating feelings of loneliness in this population. Shankar et al. [37] found that social isolation would predict lower performance in verbal fluency, immediate recall, and delayed recall in older adults’ population over a 4-year period. However, in our study, despite the fact that the deterioration difference was increased for the aforementioned tests in D2 (including assessment during lockdowns), it may be attributed to MCI as well as ADD progression rather than the quarantine effect. In sum, regarding the MCI as well as ADD populations, it cannot be concluded that any deterioration in cognitive, functional, mood and behavioral level, as measured with specific tests, is attributed to the lockdown strategies or exclusively to them. However, future studies should investigate whether similar results could be found in cognitively intact healthy adults or not.

To conclude, according to our knowledge, there are no similar studies measuring cognition, in older adults with minor and major neurocognitive disorders, longitudinally, not infected by the COVID-19 virus but have experienced a lockdown condition. Hence, no clear comparisons can be made up to now. Moreover, there are almost no studies using such a variety of neuropsychological tests, as well as psychometric tools through a three consecutive years’ time period ‘pre’ and ‘during’ the pandemic. 

Concerning mood and behavioral disturbances per se, the large corpus of literature found that mood symptoms, specifically anxiety and depression, as well as behavioral disturbances as symptoms of dementia (BPSD), are deteriorated in PwD not infected by COVID-19. However, no significant differences were found in our participants’ clinical evaluation from both groups before and during the quarantine. A possible explanation could be based on the fact that our study’s participants received telehealth support provided by our day care services, which had a possible impact on the aforementioned findings. Being able to maintain social contact during a difficult situation (imposed physical isolation due to the pandemic) may have helped them retain a good level of their psychological status. Given that social and communication platforms such as Skype, Viber and Zoom facilitate communication, significant support in older adults and mainly those with dementia is available. Recent data [38] highlight that in a significant percentage of long-term care settings across the world, caregivers were not allowed to visit their family members with dementia, and therefore, they were forced into social isolation and deprivation. Previous studies [39,40] found that social distancing is associated with medical problems such as cardiovascular disease and dementia, as well as psychiatric disorders including depression, and also increasing mortality rates. Consequently, confinement and self-isolation in patients with dementia living in nursing home facilities should be taken into account when adapting hygiene safety measures, such as keeping two meters away and isolation, due to their long-lasting outcome. Hence, the fact that we did not find differences in our study’s participants during the pandemic confinement could be attributed to the fact the participants of our study—those with MCI and ADD—were living in their homes. 

To conclude, interestingly, according to our findings, the confinement measures are not necessarily associated with inseparable increase in mood and behavioral symptoms (NPI, SAST) among older adults with MCI as well as ADD. In line with this, these symptoms during the pandemic could not predict the higher deterioration found in verbal abilities and everyday functioning as measured by the RAVLT and verbal fluency tests, and FUCAS performance, respectively, in both groups and mainly in people with ADD. Additionally, despite the fact that DSST seems to worsen in the MCI population, we could not insert it into the path model, because it was not administered in the ADD group. Future studies should identify whether the existing findings can be also implemented in cognitively intact older adults or not.

## 5. Conclusions 

Our research could be taken forward in several ways. First, our findings highlight the significance of implementing meaningful activities and psychosocial web-based interventions in older adults with MCI as well as ADD in mild and moderate stages, which is also in line with previous evidence [6,14]. Our recent study [41] was partially in line with the current results because we found that deterioration was increased in the MCI population by means of measuring their long-term memory with RAVLT as well as the phonemic verbal test. Therefore, it could be assumed that MCI participants’ performance, at least as measured by the aforementioned tests, is not affected by the quarantine. Specifically, providing valid information about the confinement measures, giving practical solutions for everyday problems, and improving the experience of being at home, can be protective factors of physical and mental health decline in older adults with MCI and ADD.

Additionally, Goodman-Casanova et al. [42] also found optimal levels of physical and mental health indices in MCI and ADD populations over a 2-week period during the quarantine in their telephone-based study. Hence, it can be assumed that the effect of lockdown either being two weeks, similarly with previous findings [42], or being 2 months, according to our analyses, does not necessarily lead to worsened psychological and physical evaluations, if structured telesupport is applied. Therefore, it could be assumed that telehealth support, either being delivered via telephone or through structured non-pharmacological interventions, can be regarded as a supporting factor during the quarantine period. This fact is further supported by the vast majority of the aforementioned studies, which found that MCI and ADD populations who did not receive this kind of support were severely affected during the lockdown measures imposed. 

Therefore, the question might then be raised as to whether longitudinal studies can shed light on the real effect of quarantine in older adults with neurocognitive disorders, in order to highlight both risk as well as protective factors which impact their cognitive as well as physical status. Thereafter, the links between the coronavirus pandemic and long-term neurocognitive and functional sequelae should be further enlightened. To conclude, research in the near future will need to investigate the long-term consequences of the COVID-19 pandemic on cognition, mood and daily functionality among older adults. 

## 6. Limitations

At first, the quality-of-life measurement was not assessed in our sample; despite the fact that Lara et al. [35] found no differences in quality of life one month before and just after the lockdown in people with MCI and ADD, it is quite important to elaborate more on this issue. Secondly, we did not include older adults who have been diagnosed with the COVID-19 virus in our sample, and therefore, they probably had a greater psychological impact from receiving a confirmed respiratory illness diagnosis. Finally, the results cannot be equally applied to cognitively healthy older adults, and therefore, they cannot be generalized to this population.

## Figures and Tables

**Figure 1 brainsci-11-01165-f001:**
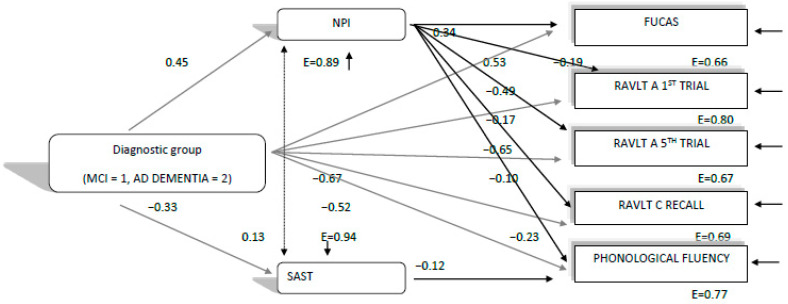
Direct and indirect relationships of diagnostic group (MCI, ADD) and negative affect (NPI, SAST) variables with everyday functioning and cognitive variables, which were found to decline faster between the 2019 and 2020 measurements than between 2018 and 2019, as measured in the 2020 neuropsychological assessment during COVID-19-related lockdowns. All relationships are significant at the *p* < 0.05 level. E = measurement error. All everyday functioning and cognitive variables were found to correlate significantly (*p* < 0.05) between each other.

**Figure 2 brainsci-11-01165-f002:**
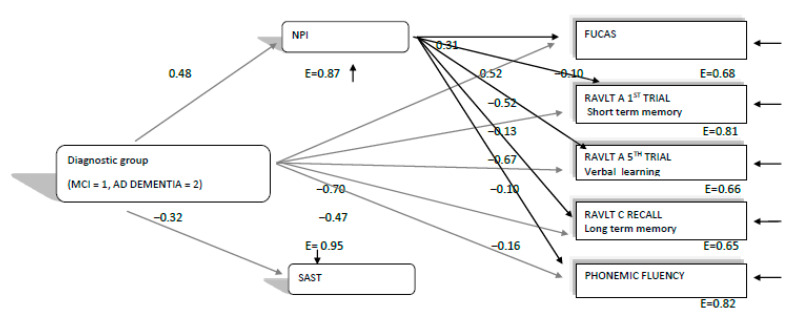
Direct and indirect relationships of diagnostic group (MCI, ADD) and negative affect (NPI, SAST) variables with everyday functioning and cognitive variables, which were found to decline faster between the 2019 and 2020 measurements than between 2018 and 2019, as measured in 2019 neuropsychological assessment. All relationships are significant at the *p* < 0.05 level. E = measurement error. All everyday functioning and cognitive variables were found to correlate significantly (*p* < 0.05) between each other.

**Figure 3 brainsci-11-01165-f003:**
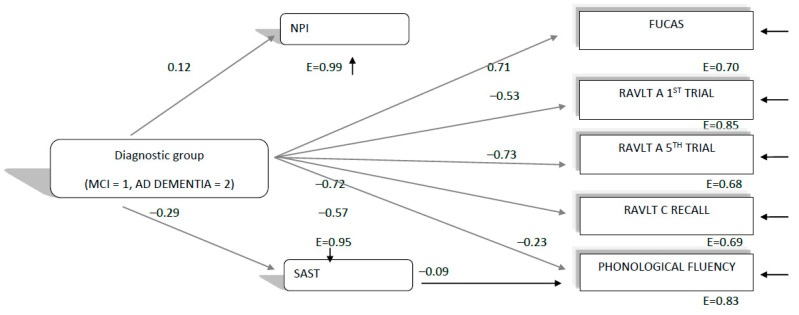
Direct and indirect relationships of diagnostic group (MCI, ADD) and negative affect (NPI, SAST) variables with everyday functioning and cognitive variables, which were found to decline faster between 2019 and 2020 measurements than between 2018 and 2019, as measured in the 2018 neuropsychological assessment. All relationships are significant at the *p* < 0.05 level. E = measurement error. All everyday functioning and cognitive variables were found to correlate significantly (*p* < 0.05) between each other.

**Table 1 brainsci-11-01165-t001:** Demographics in MCI and ADD participants.

Demographics	MCI (*n* = 296)	ADD (*n* = 111)
Gender		
Men	99	47
Women	197	64
Age (years)	71.60 (7.7)	77.32 (6.6)
Education (years)	12.38 (3.9)	9.28 (4.8)

Mild Cognitive Impairment: MCI; Alzheimer’s Disease Dementia: ADD.

**Table 2 brainsci-11-01165-t002:** Mean and standard deviation of the neuropsychological tests and mood-behavioral tests in MCI and ADD participants.

Mean (SD)					
Neuropsychological Tests/ Psychometric Tools	Group	2018	2019	2020	*p*
MMSE	MCI	27.77 (2.1)	27.73 (1.9)	27.74 (1.9)	<0.001
	ADD	21.59 (3.8)	20.53 (4.9)	19.05 (5.6)	
MoCA	MCI	23.69 (3.4)	23.77 (3.0)	23.74 (3.3)	0.550
	ADD				
FUCAS	MCI	43.77 (1.6)	43.95 (1.7)	43.75 (1.8)	<0.001
	ADD	54.20 (8.6)	55.86 (10)	60.37 (15)	
NPI	MCI	3.37 (5.9)	1.75 (7.8)	2.33 (4)	0.001
	ADD	7.27 (7.8)	7.81 (7.7)	8.78 (8.9)	
SAST	MCI	18.27 (5.1)	17.66 (4.7)	16.95 (4.7)	<0.001
	ADD	14.82 (4.5)	14.40 (3.8)	13.69 (3.3)	
RAVLT1	MCI	6.19 (2.5)	6.27 (2.3)	5.18 (1.6)	<0.001
*Short term memory*	ADD	2.89 (1.6)	2.96 (1.6)	2.65 (1.5)	
RAVLT 5	MCI	11.44 (2.4)	11.64 (2.5)	10.54 (2.5)	<0.001
*Verbal learning*	ADD	5.59 (2.2)	5.49 (2.6)	4.67 (2.5)	
RAVLT	MCI	9.58 (3.3)	9.63 (3.3)	7.71 (3.3)	<0.001
*Delayed Recall*	ADD	2.24 (2.3)	2.07 (2,1)	1.22 (1.9)	
RCFT	MCI	28.79 (4.6)	29.01 (4.7)	30.81 (3.7)	0.031
*Copy*	ADD	19.78 (4.6)	19.51 (7.9)	20.51 (10)	
RCFT	MCI	15.28 (6.1)	15.67 (6.2)	17.69 (6.9)	0.081
*Delayed Recall*	ADD	4.6 (4)	4.3 (3.8)	4.08 (4.3)	
Phonemic fluency	MCI	11.33 (3.6)	11.48 (3.6)	11.54 (3.6)	0.011
	ADD	6.34 (3)	6.35 (3.2)	5.89 (3.4)	
DSST	MCI	34.06 (12.31)	34.94 (12.13)	34.05 (12.19)	<0.001
	ADD	-			

Mild Cognitive Impairment: MCI; Alzheimer’s Disease Dementia: ADD; Standard Deviation: SD; Mini Mental State Examination: MMSE; Montreal Cognitive Assessment: MoCA; Functional Cognitive Assessment Scale: FUCAS; Neuropsychiatric Inventory: NPI; Short Anxiety Screening Test: SAST; Rey Auditory Verbal Learning Test (RAVLT); Rey Complex Figure Test: RCFT; Digit Symbol Substitution Test: DSST.

**Table 3 brainsci-11-01165-t003:** Results of the 2 × 2 mixed measures ANOVA in MCI and ADD groups across the deterioration rate levels (D1 and D2).

	Deterioration Rate (D)			Group			Group by D Interaction		
	*F*	*p*	*η2*	*F*	*p*	*η2*	*F*	*p*	*η2*
MMSE	2.06	0.152	0.007	81.5	**<0.001 ***	0.2	2.38	0.008	0.12
MoCA	0.851	0.357	0.003	------			------		
FUCAS	15.77	**<0.001 ***	0.049	103.38	**<0.001 ***	0.251	24.59	**<0.001 ***	0.317
NPI	2.67	0.103	0.009	3.7	0.055	0.012	0.76	0.383	0.003
SAST	1.16	0.281	0.004	1.15	0.136	0.001	1.35	0.245	0.004
RAVLT	23.2	**<0.001 ***	0.007	7.14	0.127	0.008	7	0.009	0.023
*First trial*									
RCFT	3.42	0.065	0.011	7.61	0.006	0.025	0.04	0.833	0.001
*Copy*									
RAVLT	24.8	**<0.001 ***	0.077	0.9	0.342	0.003	3.7	0.055	0.01
*5th trial*									
RAVLT	29.8	**<0.001 ***	0.009	3.56	0.06	0.01	10.1	**0.002 ***	0.03
*Delayed*									
RCFT	0.44	0.5	0.001	22.45	**<0.001 ***	0.07	0.19	0.658	0.001
*Delayed*									
Verbal	8.29	**0.004 ***	0.02	12.37	**0.001 ***	0.04	2.02	0.155	0.007
Fluency									
DSST	20.87	**<0.001 ***	0.09	------			------		

* Mild Cognitive Impairment: MCI; Alzheimer’s Disease Dementia: ADD; Standard Deviation: SD; Mini Mental State Examination: MMSE; Montreal Cognitive Assessment: MoCA; Functional Cognitive Assessment Scale: FUCAS; Neuropsychiatric Inventory: NPI; Short Anxiety Screening Test: SAST; Rey Auditory Verbal Learning Test (RAVLT); Rey Complex Figure Test: RCFT; Digit Symbol Substitution Test: DSST. * *p* < 0.004. Τhe separate analyses on amnestic and non-amnestic MCI were conducted and did not show differences.

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
