# Peer review of "Are There Any Cognitive and Behavioral Changes Potentially Related to Quarantine Due to the COVID-19 Pandemic in People with Mild Cognitive Impairment and AD Dementia? A Longitudinal Study"

_brainsci, 2021, doi:10.3390/brainsci11091165_

Round 1
Reviewer 1 Report
This paper is aimed at gaining further insight into relationship between cognitive decline and social distancing. In particular, the Authors formulate two hypothesis about this issue and have purpose to verify them through retrospective analysis of pre- and post-lockdown neuropsychological data.
The clinical and social issue of this work appears to be relevant at present and in the future care of patients with cognitive disorders and it is sufficiently contextualized in the introduction. Description of statistical and neuropsychological methods is exhaustive. Qualitative and quantitative findings are adequately summarized in “Results” section and the discussion is coherent with them. Even if the impact of the findings is not strongly relevant in distinguishing between natural history of disease and adjunctive effects of quarantine, this limitation seems to be correctly highlighted by the Authors. At the same time, the methodological approach of this work is quite novel and adequate for the study of dementia during pandemic social changes.
Author Response
We would like to thank you for your thoughtful and constructive comments.
Sincerely,
The authors
Reviewer 2 Report
August 6, 2021
Comments to the Author:
The current study examined the effect of the quarantine and lockdown imposed to mitigate the spread of the COVID-19 virus on cognition, emotion, and behaviours of individuals diagnosed with mild cognitive impairment (MCI) and Alzheimer’s Disease dementia (ADD). It was hypothesized that they would experience accelerated cognitive and functional decline in comparison to their pre-pandemic baseline deterioration rate prior to the pandemic, and this would be a result of emotional and behavioural changes experienced due to the lock down. The sample included 407 participants (50-92 years) from dementia care centres in Thessaloniki, Greece, who had a diagnosis of either MCI (N = 296), or ADD (N = 111). They completed a battery of neuropsychological and mood assessments in 2018, 2019, and 2020. The measures included the Mini Mental State Examination (MMSE), the Montreal Cognitive Assessment (MoCA), components of the Rey’s Verbal Learning Test (RAVLT; short term memory, verbal fluency, delayed recall measure long term memory), a phonemic verbal fluency test, Rey’s Complex Figure Test (RCFT), the Wechsler Adult Intelligence Scale- Digit Symbol Substitution Test (WAIS-DSST), the Functional Cognitive Assessment Scale (FUCAS), the Short Anxiety Screening Test (SAST), and the Neuropsychiatric Inventory (NPI). All measures were validated for the Greek population. A mixed-model 2 (group: MCI compared to AD) x 2 (deterioration: D1 deterioration between 2018 to 2019 vs. D2 deterioration between 2019 to 2020) ANOVA was conducted on the participants’ scores from the aforementioned neuropsychological tests and psychometric tools to assess group difference in deterioration rate prior and during the pandemic. A path analysis was conducted to understand the relationships between mood variables and cognitive and everyday functioning. Both groups did not show significant change in mood or behaviour deterioration, however, there was a larger deterioration during the pandemic (from 2019-2020) in FUCAS, RAVLT, and phonemic fluency, as compared to baseline (from 2018 to 2019). The path analysis showed that NPI predicted decline in performance on the FUCAS, RAVLT, and phonemic fluency in 2019 and 2020, but not in 2018. A comprehensive battery of measures were employed and the study has a longitudinal component. But there are some methodological and discussion concerns that make interpretation of the results questionable.
Main comments
- As the main independent variable, lockdown or quarantine experience, was neither measured or manipulated in this study. Thus the conclusion that the steeper deterioration during the pandemic is drive by the lockdown/quarantine is largely speculative. It was unclear to what degree the participants adhered to the lockdown/quarantine measures, and if so how long they had done so (e.g., was their face-to-face assessment in 2020 the first outing they had in two weeks, or a longer duration?). Additionally, some critical information related to home confinement, social isolation, and decreased access to health/social support a is missing, such as living situation (e.g., private home or long-term care facility; alone or with others), telehealth care service frequency and nature (e.g., medical or psychosocial intervention). Without this information, it is not guaranteed that the results of the current study are, as being claimed, associated with the COVID-19 quarantine and lockdown. The accelerated deterioration might be just the reflection of the natural progress of the condition, regardless of the lockdown or pandemic.
- In a related note, without a healthy control group, it is also uncertain whether the results are specific to MCI and ADD, or equally apply to health older adults.
- The introduction could benefit from some theoretical or practical rationales to stress the unique contribution of the current study. It was mentioned that previous research has not used neuropsychological tests in the “Discussion” section (line 435-436), which might nicely fit in the introduction. The paragraph (line 492-497) would serve as a transition into the aims of the study in the introduction.
- Line 112, the sample consisted of people with amnestic and non-amnestic MCI. Would this be a confound in some cognitive performance (e.g., delayed recall)? What the results would look like if they were analyzed as separate groups? Please clarify.
- The ADD participants did not complete MoCA and DSST. Please provide a brief justification.
- The authors used the SAST to measure mood deficits. This screener is valid in assessing anxiety in the presence of depression in older adults, but has it been validated for those with MCI or ADD? As the NPI also assesses depressive and anxiety symptoms, what did the SAST add to the study?
- The results were discussed in light of the progression of ADD but less of MCI. It makes sense that the decline was more pronounced in ADD than MCI. It will be valuable to further discuss the MCI symptomology progression as well.
- The MCI and ADD groups differed in age and education, how these might contribute to your results? Would it be possible to address this in your data analysis and discussion?
- Please discuss the key findings of Goodman-Casanova et al. (2020) in a way that is meaningful for the conclusion that the duration of the lockdown may not lead to worsening psychological or physical symptoms unless certain conditions are met? What contributes to the optimal levels of physical/mental indices for MCI and ADD?
- Defining acronyms was inconsistent throughout the document. For example, ADD appears as AD (line 458), Alzheimer’s disease (line 488). Acronyms appear before their definition, and some were incorrect. BPSD first appears in line 93, then line 189, and is finally defined as “behavioural disturbances as symptoms of dementia” in like 449-450; instead of “behavioural and psychiatric symptoms of dementia”? Please be consistent.
Minor comments: There are some typographical errors in style, grammar, and run-on sentences etc., to name a few:
- Line 113- Do not use the roman numeral for DSM-5.
- Line 98- Avoid using first person point of view. Instead of “We aimed…” try, “The researchers aimed…”
- Line 150- “In specific…” should be “Specifically…”
- Line 213, 223, 274, 275, 318- Should be “in regard to…” or “regarding…” rather than “as regards…”
- Line 422, 423- Did you mean, “did not significantly differ…”?
- Line 427- the word “marginally” in brackets should be replaced with what the phonemic verbal fluency test measures to be consistent with what is in brackets following the tests in the same sentence.
- Line 516- you mention “hygiene safety,” are you trying to describe COVID-19 protective measures such as keeping two metres away and isolating? Or hand washing, wearing a mask?
- Line 522- In brackets you should have (NPI, SAST), BPSD is what the NPI measures.
- Line 522- Should be “could not” rather than “couldn’t.”
- In the “Screening Tests” section of the results, for MMSE and MoCA, “S.E.” was reported instead of SD? please justify.
- Line 48- The Diagnostic and Statistical Manual of Mental Disorders (DSM-5) lists MCI as Mild Neurocognitive Disorder (can be found on page 605 in the DSM-5). Within the manuscript reads “Minor Neurocognitive Disorders” (line 148).
- In line 482, the authors mention that the lockdown condition would lead to decreasing interest to communicate. Would it not lead to increased interest to communicate, and the inability to do so exacerbating feelings of loneliness in individuals?
- Please specify the exact time frame for the administration of the assessment instead of the vague description of “the same three months’ period” (line 117). This is particularly important in the context of COVID-19 research, as the experience of the pandemic was quite dynamic month to month in 2020.
Author Response
First of all, we would like to thank you for your thoughtful and constructive comments. We have been working on the revisions and we believe that the quality of our manuscript has greatly improved with the incorporation of your comments. In the following document you will find our answers to your comments.
Sincerely,
The authors
Main comments
- As the main independent variable, lockdown or quarantine experience, was neither measured or manipulated in this study. Thus the conclusion that the steeper deterioration during the pandemic is drive by the lockdown/quarantine is largely speculative. It was unclear to what degree the participants adhered to the lockdown/quarantine measures, and if so how long they had done so (e.g., was their face-to-face assessment in 2020 the first outing they had in two weeks, or a longer duration?). Additionally, some critical information related to home confinement, social isolation, and decreased access to health/social support a is missing, such as living situation (e.g., private home or long-term care facility; alone or with others), telehealth care service frequency and nature (e.g., medical or psychosocial intervention). Without this information, it is not guaranteed that the results of the current study are, as being claimed, associated with the COVID-19 quarantine and lockdown. The accelerated deterioration might be just the reflection of the natural progress of the condition, regardless of the lockdown or pandemic.
We want to thank Reviewer 2 for this comment. Actually, the independent variable of the sample was diagnosis, because quarantine experience was not spread through the three years duration of our study. In the conclusion, we state that even the deterioration in the three tests’ performance cant be attributed to the quarantine, but in disease’s progress.
Concerning the degree in which participants adhered to the lock down measures, as Reviewer 2 said, we couldn’t control in which level participants compromise with the safety measures, but in general there were national guidelines imposed for all Greek citizens.
‘Additionally, some critical information related to home confinement, social isolation, and decreased access to health/social support a is missing, such as living situation (e.g., private home or long-term care facility; alone or with others), telehealth care service frequency and nature (e.g., medical or psychosocial intervention).’ Reviewer 2 is right. All these could be very significant information, however, our study would be enlarged and therefore difficult for the reader to follow.. However, we are now preparing the sequel study in which we gain information about these variables.
‘The accelerated deterioration might be just the reflection of the natural progress of the condition, regardless of the lockdown or pandemic.’ Reviewer 2 is right. We claim that the natural progress of the condition is the main reason for participants’ deterioration.
- In a related note, without a healthy control group, it is also uncertain whether the results are specific to MCI and ADD, or equally apply to health older adults.
In our sample we had very few healthy controls and therefore, no significant results could be gained for this population. Therefore, the results are attributed in MCI and ADD participants , according to the this study.
- The introduction could benefit from some theoretical or practical rationales to stress the unique contribution of the current study. It was mentioned that previous research has not used neuropsychological tests in the “Discussion” section (line 435-436), which might nicely fit in the introduction. The paragraph (line 92-97) would serve as a transition into the aims of the study in the introduction.
We want to thank Reviewer 2 for this comment. Please see the relevant text in lines 96-100 in yellow font.
- Line 112, the sample consisted of people with amnestic and non-amnestic MCI. Would this be a confound in some cognitive performance (e.g., delayed recall)? What the results would look like if they were analyzed as separate groups? Please clarify.
Very interesting comment. We did separate analyses, but no differences were observed. We just tried to reduce the length of our study, this is the reason why we didn’t report the extra analyses.
- The ADD participants did not complete MoCA and DSST. Please provide a brief justification.
MoCA and DSST tests are not administered in ADD participants, mainly those in moderate and severe stages, because they can’t respond to the tests’ demands.
- The authors used the SAST to measure mood deficits. This screener is valid in assessing anxiety in the presence of depression in older adults, but has it been validated for those with MCI or ADD? As the NPI also assesses depressive and anxiety symptoms, what did the SAST add to the study?
Thank you for this comment. We used SAST in order to receive participants’ response about their anxiety levels complementary to the NPI scores. The same for GDS. Unfortunately, there are no study in Greek for SAST validation in MCI and ADD populations, however, according to our clinical experience, we use these tests in our daily practice.
- The results were discussed in light of the progression of ADD but less of MCI. It makes sense that the decline was more pronounced in ADD than MCI. It will be valuable to further discuss the MCI symptomology progression as well.
The reason why we focused on ADD rather than MCI is because when comparing the two groups, ADD participants were vastly impaired, as expected, compared to the MCI. In the majority of tests, MCI seem to have no further impairment and therefore we tried to explain the deterioration found in ADD participants.
- The MCI and ADD groups differed in age and education, how these might contribute to your results? Would it be possible to address this in your data analysis and discussion?
ADD participants are usually older and less educated compared to the MCI ones. In our study we didn’t find any differences due to the pandemic, this is the reason why we didn’t proceed to relevant explanation regarding demographics.
- Please discuss the key findings of Goodman-Casanova et al. (2020) in a way that is meaningful for the conclusion that the duration of the lockdown may not lead to worsening psychological or physical symptoms unless certain conditions are met? What contributes to the optimal levels of physical/mental indices for MCI and ADD?
We want to thank Reviewer 2 for this comment. Please see the lines 548-553 in yellow font.
- Defining acronyms was inconsistent throughout the document. For example, ADD appears as AD (line 458), Alzheimer’s disease (line 488). Acronyms appear before their definition, and some were incorrect. BPSD first appears in line 93, then line 189, and is finally defined as “behavioural disturbances as symptoms of dementia” in like 449-450; instead of “behavioural and psychiatric symptoms of dementia”? Please be consistent.
We have now used the term neuropsychiatric symptoms’ in the aforementioned definitions.
Minor comments: There are some typographical errors in style, grammar, and run-on sentences etc., to name a few:
- Line 113- Do not use the roman numeral for DSM-5.
Done
- Line 98- Avoid using first person point of view. Instead of “We aimed…” try, “The researchers aimed…”
Done
- Line 150- “In specific…” should be “Specifically…”
Done
- Line 213, 223, 274, 275, 318- Should be “in regard to…” or “regarding…” rather than “as regards…”
Done
- Line 422, 423- Did you mean, “did not significantly differ…”?
Yes, we added the word ‘significantly’
- Line 427- the word “marginally” in brackets should be replaced with what the phonemic verbal fluency test measures to be consistent with what is in brackets following the tests in the same sentence.
Done
- Line 516- you mention “hygiene safety,” are you trying to describe COVID-19 protective measures such as keeping two metres away and isolating? Or hand washing, wearing a mask?
We are trying to describe protective measures such as keeping two metres away and isolating (it has been added in the text)
- Line 522- In brackets you should have (NPI, SAST), BPSD is what the NPI measures.
Done
- Line 522- Should be “could not” rather than “couldn’t.”
Done
- In the “Screening Tests” section of the results, for MMSE and MoCA, “S.E.” was reported instead of SD? please justify.
Reviewer 2 is right. SE was reported rather than SD
- Line 48- The Diagnostic and Statistical Manual of Mental Disorders (DSM-5) lists MCI as Mild Neurocognitive Disorder (can be found on page 605 in the DSM-5). Within the manuscript reads “Minor Neurocognitive Disorders” (line 148).
Done
- In line 482, the authors mention that the lockdown condition would lead to decreasing interest to communicate. Would it not lead to increased interest to communicate, and the inability to do so exacerbating feelings of loneliness in individuals?
It has been now added
- Please specify the exact time frame for the administration of the assessment instead of the vague description of “the same three months’ period” (line 117). This is particularly important in the context of COVID-19 research, as the experience of the pandemic was quite dynamic month to month in 2020.
The three months’ period was from May to July , it has been now added in the text

Round 2
Reviewer 2 Report
Overall, it looks like the authors have tried to address most of the comments raised in my review. The writing has also improved in the revision. However, there are a few major comments need to be further and consistently addressed throughout the document in the revision. Please see below:
- To address the first major comment on the contribution of quarantine, the authors did agree with this comment and stated in the response letter that “We claim that the natural progress of the condition is the main reason for participants’ deterioration.” but this is rarely reflected in the revised manuscript. For example, quarantine or lockdown is still mentioned as the main variable in the title, first sentence and last sentence of the abstract. It is OK to speculate the role of quarantine in discussion, but without the controlled manipulation, these statements might indicate an over-statement of the results. Please address this throughout the document
- The second item was about whether the results would equally apply to healthy older adults. It would be beneficial to always describe the conclusion in the context specific to MCI and ADD to avoid over-generalization of the results. You may also list this (no health control) as a limitation in the discussion.
- The addition of the rationale (line 96 – 100) could be strengthened by some specific justifications. Lack of previous work does not justify it is a meaningful question. Please specify why this is meaningful (theoretically or practically)
- It is nice to learn that the separate analysis on amnestic and non-amnestic MCI were conducted and did not show differences, but his worth a mention in the results or as a footnote.
- Line 547-552: Are the conditions that would result in optimal levels of physical and mental health for those with MCI and ADD attributed to only the telehealth intervention in Goodman-Casanova et al.’s (2020) study? The reader might still be unclear about what the authors mean when they write “if certain conditions are met”? please be specific.
- ADD and AD (or Alzheimer’s disease) are still all present in the revision, please be consistent throughout the document to minimize confusion (e.g., line 568).
- In the “Screening tests” section, please justify why SE instead of SD was reported?
Some more items for further consideration:
- Line 140: “In more specific..” should be “More specifically…” or “To be more specific…”
- Line 117, 155 & 156, 161 & 161: At first mention, define the Diagnostic and Statistical Manual of Mental Disorders, fifth edition (DSM-5; line 117). Further mention of the manual does not need to be spelled out fully, as you have it in your inclusion criteria. For example, line 155 & 156 could read as, “a) a diagnosis of MCI according to the DSM-5…”
- Line 454: BPSD is not defined in manuscript after editing. Can define at this line.
Author Response
We want to thank Reviewer 2 for the valuable comments which improved our work. Please find our responses below:
- To address the first major comment on the contribution of quarantine, the authors did agree with this comment and stated in the response letter that “We claim that the natural progress of the condition is the main reason for participants’ deterioration.” but this is rarely reflected in the revised manuscript. For example, quarantine or lockdown is still mentioned as the main variable in the title, first sentence and last sentence of the abstract. It is OK to speculate the role of quarantine in discussion, but without the controlled manipulation, these statements might indicate an over-statement of the results. Please address this throughout the document
Thank you for repeating this comment, we hope that now it is even more clear.
(please see the yellow font in Abstract, lines 446-449, 479-484, 509-516)
- The second item was about whether the results would equally apply to healthy older adults. It would be beneficial to always describe the conclusion in the context specific to MCI and ADD to avoid over-generalization of the results. You may also list this (no health control) as a limitation in the discussion.
A new sentence has been not placed in the second paragraph of the Discussion (please see the yellow font), lines 505-509, 546-548, and also in the limitations’ section.
- The addition of the rationale (line 96 – 100) could be strengthened by some specific justifications. Lack of previous work does not justify it is a meaningful question. Please specify why this is meaningful (theoretically or practically)
We have now provided relevant justification
- It is nice to learn that the separate analysis on amnestic and non-amnestic MCI were conducted and did not show differences, but his worth a mention in the results or as a footnote.
We have added this statement under the Table 3
- Line 547-552: Are the conditions that would result in optimal levels of physical and mental health for those with MCI and ADD attributed to only the telehealth intervention in Goodman-Casanova et al.’s (2020) study? The reader might still be unclear about what the authors mean when they write “if certain conditions are met”? please be specific.
Thank you for this comment, we have now changed it ‘if structured tele support is applied’
- ADD and AD (or Alzheimer’s disease) are still all present in the revision, please be consistent throughout the document to minimize confusion (e.g., line 568).
Done
- In the “Screening tests” section, please justify why SE instead of SD was reported?
The reason for this report is because the SPSS output gave us the SE in the relevant analyses.
Some more items for further consideration:
- Line 140: “In more specific..” should be “More specifically…” or “To be more specific…”
Done
- Line 117, 155 & 156, 161 & 161: At first mention, define the Diagnostic and Statistical Manual of Mental Disorders, fifth edition (DSM-5; line 117). Further mention of the manual does not need to be spelled out fully, as you have it in your inclusion criteria. For example, line 155 & 156 could read as, “a) a diagnosis of MCI according to the DSM-5…”
Done
- Line 454: BPSD is not defined in manuscript after editing. Can define at this line.
Done